# Development of a 3D Eulerian/Lagrangian Aircraft Icing Simulation Solver Based on OpenFOAM

**DOI:** 10.3390/e24101365

**Published:** 2022-09-27

**Authors:** Han Han, Zifei Yin, Yijun Ning, Hong Liu

**Affiliations:** 1School of Aeronautics and Astronautics, Shanghai Jiao Tong University, Shanghai 200240, China; 2Shenyang Key Laboratory of Aircraft Icing and Ice Protection, Shenyang 110034, China

**Keywords:** icing simulation, supercooled large droplet, OpenFOAM, hybrid droplet tracking, 3D solver

## Abstract

A 3D icing simulation code is developed in the open-source CFD toolbox OpenFOAM. A hybrid Cartesian/body-fitted meshing method is used to generate high-quality meshes around complex ice shapes. Steady-state 3D Reynolds-averaged Navier-Stokes (RANS) equations are solved to provide the ensemble-averaged flow around the airfoil. Considering the multi-scale nature of droplet size distribution, and more importantly, to represent the less uniform nature of the Super-cooled Large Droplets (SLD), two droplet tracking methods are realized: the Eulerian method is used to track the small-size droplets (below 50 μm) for the sake of efficiency; the Lagrangian method with random sampling is used to track the large droplets (above 50 μm); the heat transfer of the surface overflow is solved on a virtual surface mesh; the ice accumulation is estimated via the Myers model; finally, the final ice shape is predicted by time marching. Limited by the availability of experimental data, validations are performed on 3D simulations of 2D geometries using the Eulerian and Lagrangian methods, respectively. The code proves to be feasible and accurate enough in predicting ice shapes. Finally, an icing simulation result of the M6 wing is presented to illustrate the full 3D capability.

## 1. Introduction

When an aircraft flies through a low-temperature cloud that is rich in moisture, the supercooled droplets in the cloud will impinge the aircraft surface, probably accompanied by ice accretion through mass and heat exchange phenomenon [1]. The aircraft icing can sometimes affect aerodynamic performances to a great extent like lift reduction and drag increase and thus lead to flight out of control, and sometimes detrimental accidents [2]. Efforts have been made to prevent ice from accumulating on aircraft wings or turbines. Considering that wind tunnel measurements and in-flight tests are costly and sometimes self-limited on account of testing devices, computational simulation tools are developed to predict ice shapes and perform anti-icing simulations. As a result, many numerical icing simulation codes have been developed to predict ice shapes, including LEWICE [3], FENSAP-ICE [4], the work by ONERA [5], etc.

In the last two decades, 2D and 2.5D in-flight ice prediction codes have been developed and widely used by the aerospace industry. Sometimes, wind tunnel tests can engage scaling issues if sub-scale models are used or blockage and configuration problems if full-scale models are used where full configuration and full-scale effects are important. However, the 2D numerical methods cannot be employed on 3D geometries such as multi-element wings, nacelles, or engines, as well as integrated systems or situations that combine external and internal flows. Obviously 2D icing simulation does not take into account corrections for the sweep, up-wash, down-wash, twist, etc., which may lead to inaccuracy in predicting ice shapes.

Many 3D icing simulation codes are costly and complicated, mainly in generating 3D mesh and obtaining the airflow solution, tracking droplets’ trajectories, and calculating ice accretion. For mesh generating, it has been revealed that high-quality mesh deformation techniques are essential to achieve a robust, accurate, and efficient process of flux calculations, especially viscous fluxes near the solid wall boundaries [6]. The demand for mesh moving and stretching is obvious when ice accretion occurs on the aircraft surface. However, the moving-mesh methods are quite computationally expensive. For tracking droplets’ trajectories, there are two kinds of water droplet collection methods that are both used in icing simulation codes: The Eulerian method and the Lagrangian method. Classical icing codes such as FENSAP-ICE [4] use the Eulerian method; LEWICE [3] and ONERA [5] use the Lagrangian method. The Eulerian method treats the particle phase as a continuum and develops its conservation equations on a control volume basis. The Lagrangian method considers the particle as a discrete phase and traces the pathway of each individual particle. The two methods are usually used to solve different problems. The Eulerian method proves to have good performance when dealing with concentrated and uniform distributions [7,8,9]. On the contrary, the Lagrangian method is mainly used to predict the overall particle dispersion pattern [10]. Comparatively speaking, the Eulerian method is more efficient than the Lagrangian method in predicting concentrated distributions under steady-state conditions [11].

Furthermore, the traditional difficulties encountered by the Lagrangian method include long calculation times, difficulty in determining impingement limits, sparsely seeded particles not covering complex surfaces, etc. For the ice accretion model, the traditional Messinger model [12] lacks the ability to calculate the heat transfer of the interface between the ice and water layers and cannot transition smoothly from rime ice to glaze ice when the calculation is performed. As an extension to the Messinger model, the Myers model [13] not only catches the transient conduction through the ice and water layers but also allows the complex three-dimensional water flow. Thus, the Myers model is more advantageous than the Messinger model in predicting glaze ice and mixed ice.

This paper aims to provide a 3D icing simulation code implemented in OpenFOAM [14], which is quite advantageous in mesh generation, droplet collection method, and ice accretion. This paper is structured as follows: Section 2 introduces the overall structure of the icing simulation code and the numerical methods and physical models; Section 3 demonstrates the 2.5D validation results to prove the calculation accuracy; Section 4 shows a 3D icing simulation result of the M6 wing.

## 2. Numerical Methods and Physical Models of Icing Simulation

The whole process of the icing simulation is based on time marching to accumulate ice. Within each time step, the following four steps are executed sequentially: (1) calculation of the aerodynamic flow field, by neglecting droplets’ effect on the airflow; (2) tracking the droplets’ trajectories due to aerodynamic force and computing the collected water mass on the aircraft surface; (3) an inner loop to add the droplet mass to the water film, solve the water film flow, and then estimate ice accumulation via heat transfer analysis; (4) generating the body-fitted mesh with ice accretion covered on the surface. The relation between the four steps and the outer time step loop is shown in Figure 1. The above procedure is implemented into the open-source CFD toolbox OpenFOAM [15]. Each step has its own independent solver written in C++ to realize the functionality mentioned above. A Python script serves as an outer loop (time step) driver to enable the whole automatic solution process and handle the inputs and outputs.

Before diving into the numerical methods and physical models, we want to emphasize that aircraft icing, which involves complex flow, heat transfer, and phase change, is intrinsically a complicated thermodynamic phenomenon. The simulated physical process that results in ice accumulation, is irreversible. For example, the PDEs we solve for the air and water film flow, are convection-diffusion equations. The viscosity and diffusivity in the momentum and heat equations are always positive. Obviously, one can not solve the convection-diffusion phenomenon backward in time. In addition to the PDEs, some approximations and empirical models applied are also irreversible: in the Lagrange tracking method, a droplet that has a Weber number higher than a certain threshold is subject to splashing when approaching airfoil surface; in the icing model, the kinetic energy of the flying droplets is converted into the thermal energy of the water film; the Myers model, which provides an empirical solution to the 1D Stefan problem along the wall normal direction, is also irreversible in time. Overall, the nature and computer simulation of ice accumulation over time, is directed by the second law of thermodynamics, which cannot be reversed.

### 2.1. Mesh Generation

Traditional finite-difference and finite-volume CFD methods rely on a structured body-fitted mesh to accurately compute the near-wall gradients. Indeed, body-fitted mesh allows accurate computation of wall stresses and heat fluxes. Note that the mesh has to be regenerated at each icing step due to the change of boundary shape which requires an automatic mesh generation method. Methods using structures mesh can stretch the mesh to fit the ice shape while keeping the topology unchanged. However, 3D ice accumulation can become fairly irregular, and significant deterioration of near wall mesh quality can be observed. Artificial smoothing on the ice shape may be necessary for structured mesh, which will certainly sacrifice the accuracy on the predicted ice shape for the need of computation robustness.

To overcome this, a mixed Cartesian/body-fitting mesh, as shown in Figure 2 is used. The mixed Cartesian/body-fitting approach balances well between the complex ice shape and the need for computing wall stresses/fluxes. The Cartesian mesh in the background with local refinement allow a better geometric representation of the ice shape without severe deterioration of the mesh quality and a general reduction in total cell number. Furthermore, additional body-fitted anisotropic wall layers enable the calculation of variables such as the velocity very close to the wall with comparatively few additional grid cells, which ensures a precise evaluation of the aerodynamic forces and heat transfer. The resulting $y_{+}$ is around $10^{2}$ which is located inside the log layer for a typical turbulent boundary layer. It is worth noticing that some key regions like the leading edge where the ice accretion is mostly likely to occur need to have extra refinement to ensure the ability to capture the ice shape accurately. Finally, body-fitted layers are added to ensure the precise evaluation of the aerodynamic forces and heat transfer. The above meshing technique is available in snappyHexMesh [16] where user-defined parameters are needed for the automatic generation of high-quality mesh.

The mesh convergence study on the lift coefficient was performed for the NACA 0012 airfoil at the AOA of $4^{\circ }$ with a comparison made to experimental data from McCroskey [17]. The simulation is performed in 2D. The numerical setup for aerodynamic simulation is introduced in Section 2.2. Note that, unlike the traditional O-type or C-type structured meshes, there are many free parameters to adjust for snappyHexMesh. The ranges of some tested parameters are shown in Table 1. The starting point is with a Cartesian refinement level of 3, size of refinement zone of 1.5c×0.5c, number of wall layers of 7, and farfield distance of 10c. The effect of each parameter is explored individually with the error measured on the lift coefficient. The resulting mesh after the sensitivity study, is adjusted to have a farfield distance of 25c. There are 10 wall layers near the airfoil surface. The initial uniform background mesh has a size of 50×50 before the Cartesian cuts. The automatic refinement level near the airfoil surface ranges from 7 to 9. The interior refinement zone is confined within the distance of 0.25c to the airfoil surface to keep down the total cell size. The resulting lift coefficient error is about 1.2%, which is a trade-off between efficiency and accuracy.

One may point out that traditional mesh sensitivity study on aerodynamic simulations involves $C_{p}$, $C_{f}$, $C_{l}$, $C_{d}$, and $C_{m}$. Note that even the famous LEWICE [3] icing solver uses a potential solver to estimate the airflow, yet it is still one of the most successful and trustworthy icing solvers in the world. In the current icing simulation, the aerodynamic field near the leading edge is the most important, which is the key for predicting the droplets’ impingement. The trailing edge wake, however, has little influence on the ice shape. It is known that the numerical resolution of the trailing edge wake has a significant influence on the drag coefficient. We decide to sacrifice the mesh resolution near the trailing edge in order to gain some efficiency in icing simulation. That is why our mesh sensitivity study is only measured using the lift coefficient.

According to the authors’ experience, the water film is less sensitive to the grid resolution. Due to the extra refinement of Cartesian mesh within the potential icing region, the ice shape does not show much sensitivity to the grid resolution.

### 2.2. Solution of the Airflow Field

Due to the mass, momentum, and heat exchange between the air and ice/film, the aerodynamic and temperature fields of the flow around the airfoil are prerequisites for solving ice accretion. The thickness of the water film is neglected here because of its insignificance relative to the airfoil size. Because the icing time scale can be in minutes which is significantly larger than the flow (turbulence) time scale, thus the airflow is assumed to be steady within each icing step. It is a standard practice in many icing software. The steady solution of the ensemble-averaged airflow field around the airfoil is obtained by solving Reynolds-averaged Navier-Stokes (RANS) equations in OpenFOAM. The compressible Semi-Implicit Method for Pressure Linked Equations (SIMPLE) [18] method is used to solve the pressure-velocity coupling (via solver rhoSimpleFoam). Generally for external aerodynamics, the Spalart-Allmaras model [19] and Menter’s k−ω shear stress transport (SST) model [20] are popular. In the present case, the flow can be separated due to irregular ice shape, thus the Menter’s k−ω shear stress transport (SST) model is chosen for the following simulations. The Prandtl number and the turbulent Prandtl number of air is set to 0.71 and 0.9, respectively. The “all $y_{+}$” wall function is adopted to ensure proper computation of velocity and temperature field. All the transport equations are solved using the preconditioned Bi-conjugate gradient method, and the pressure equation is solved by the preconditioned conjugate gradient method.

In the airflow simulations, Dirichlet boundary condition is given for velocity, turbulent kinetic energy, eddy frequency, and temperature at the inlet. Zero gradient boundary condition is applied at outflow for the transported variables. Pressure is set to zero gradient at inflow and fixed at outflow. To accelerate convergence, the velocity field is initialized by a potential solver.

Considering the mesh can be non-orthogonal, for the purpose of solver robustness, the gradients are evaluated using the least-squares approach and are limited to ensure that the face values obtained by extrapolating the cell value to the cell faces using the gradient are bounded by the neighboring cells minimum and maximum limits. The convection of velocity is discretized by a second-order linear upwind scheme, with an explicit correction based on the local cell gradient. The convection of turbulent kinetic energy and eddy frequency are discretized by the upwind scheme to ensure boundedness. The number of non-orthogonal corrections is set to 2 to account for the max mesh non-orthogonality no greater than 65.

For clean airfoil and rime ice conditions, where no significant flow separation is present, the velocity residual converges below $10^{-6}$ while the pressure (continuity) residual is about $10^{-4}$. In the case of flow separation caused by ice shape, the pressure (continuity) residual can be up to $10^{-3}$. The authors have performed a mesh convergence study on the NACA 0012 airfoil. The current meshing parameters and the numerical setup guarantee accurate lift and drag coefficients prediction before stall, given that the airfoil shape is discretized with more than 200 faces–a standard practice in many external aerodynamics simulations. Thus, the flow simulation is considered trustworthy.

### 2.3. Droplet Tracking and Collecting Method

The method is established based on the following assumptions: (1) droplets do not collide with one another and they do not have mass, momentum, or heat exchange; (2) droplets are deemed as spheres in calculating aerodynamic forces; (3) large droplets may experience break-up before impacting the surface; (4) splashing is only considered for the large droplets; (5) only the drag, gravity, and buoyancy are considered as the forces of airflow exerted on droplets; (6) the effect of droplets on airflow is negligible. These assumptions are applicable under the condition that the volumetric fraction of droplets is low (no more than $10^{-6}$).

A large number of small droplets (below 50 μm) in the cloud can be regarded as uniformly distributed, thus an Eulerian method seems appropriate; The SLDs are lesser and their diameters and locations may vary significantly. Thus using the Lagrangian method to track their location and to represent their breakup and splashing is more accurate.

#### 2.3.1. The Eulerian Method

As mentioned, the small-size droplets can be viewed as a continuous phase. Moreover, the Eulerian method is more efficient than the Lagrangian method in solving concentration problems under steady-state conditions. Thus the Eulerian method is more suitable in calculating the collection of small-size droplets. The Eulerian formulation of the droplet impingement process yields a set of droplet-related continuity and momentum equations, which are introduced by Bourgault et al. [21]. Since $\rho_{d}$ is constant, the transport equations are defined as follows,
(1)$$\frac{\partial \alpha }{\partial t}+\nabla \cdot \left(\alpha U_{d}\right)=0,$$
(2)$$\frac{\partial \left(\alpha U_{d}\right)}{\partial t}+\nabla \left(\alpha U⊗U_{d}\right)=\frac{\alpha C_{D}Re_{d}}{24K}\left(U_{a}-U_{d}\right)+\left(1-\frac{\rho_{a}}{\rho_{d}}\right)\frac{\alpha g}{Fr^{2}}.$$
In Equation (Equation 1), $\alpha =LWC/\rho_{d}$ represents the non-dimensionalized volumetric fraction of droplets, where LWC is the liquid water content, as in kg/m${}^{3}$. In Equation (Equation 2), $U_{a}$ and $U_{d}$ are velocities of the airflow and droplets. $K=\rho_{d}d^{2}/18L\mu$; $C_{D}=\left(24/Re_{d}\right)\left(1+0.015{Re_{d}}^{0.687}\right)$ for $Re_{d}\leq$ 1000, and $C_{D}=0.4$ for $Re_{d}>$ 1000 represents the drag coefficient for sphere droplets; $Re_{d}=\rho_{d}d\left|U_{a}-U_{d}\right|/\mu$ represents the droplets’ Reynolds number; $F_{r}=U_{\infty }/\sqrt{Lg}$ represents the Froude number; $\rho_{d}$ represents the density of water; *d* represents the droplet diameter, usually chosen equal to the mean volume diameter (MVD) of the droplet size distribution; *L* represents the characteristic length (typically the airfoil chord length); μ is the dynamic viscosity at air. The droplet momentum in Equation (Equation 2) is expressed in a non-conservative form and formulated based on the assumption that a droplet in motion is only subject to the aerodynamic drag, buoyancy and gravity forces. The first right-hand-side term of Equation (Equation 2) represents the air drag force on the droplets, and the second term represents the buoyancy and gravity forces.

Under the icing conditions, it is considered that the airflow and the droplet are one-way coupled, namely the airflow field affects the droplet field only since the volumetric fraction of droplets is too low (usually below $10^{-6}$) to make the droplet field affect the airflow field comparatively [22], which means that the airflow field and the droplet field can be solved separately: in each iteration step, the solution of the airflow field is first solved, and then the result is used to solve the droplet field.

The collection efficiency is then defined as
(3)$$\beta =\frac{\alpha_{n}\left(U_{n}\cdot n\right)}{\alpha_{\infty }U_{\infty }},$$
where $U_{n}$ represents the droplet velocity at the cell center nearest to wall; n represents the normal unit vector of the airfoil surface; $\alpha_{n}$ represents the volumetric fraction of droplets at the wall boundary field.

#### 2.3.2. The Lagrangian Method

SLDs are defined as droplets whose sizes are over 50 μm. They are better treated in a Lagrangian way because the physical properties of SLDs differ from conventional small-size droplets. For instance, SLDs are distributed more sparsely than small-size droplets, thus an Eulerian approach may not be accurate; SLDs are more difficult to change their trajectories because of their greater inertia, and therefore more likely to collide with the aircraft; also, SLDs are prone to splashing [23], which makes their tracking complicated and requires additional physical models.

In the current implementation, the large droplets are tracked using parcels, as a collection of droplets with the same physical properties. Sensitivity analysis is performed so that the solution is independent of the total number of parcels (no less than $10^{4}$). The Lagrangian governing equation of each parcel is
(4)$$\frac{dU_{d}}{dt}=\frac{C_{D}Re_{d}}{24K}\left(U_{a}-U_{d}\right)+\left(1-\frac{\rho_{a}}{\rho_{d}}\right)g.$$
The right-hand-side terms and variables of Equation (Equation 4) share the same meaning with Equation (Equation 2).

Numerically, a set of ODEs (Equation (Equation 4)) is marched via fourth-order Runge-Kutta method. Moreover, a method of locating a droplet accurately is needed by which Barycentric interpolation can be performed in order to move the particle through the domain and to obtain $U_{a}$ at any position [24]. The method avoids searching for the particle inside the entire domain, which significantly improves the solver efficiency.

After the droplet tracking process, the collection efficiency is then defined as:(5)$$\beta =\frac{M_{collected}}{LWC\cdot u\cdot \left|S\right|\cdot t}$$
In Equation (Equation 5), $M_{collected}$ represents the collected mass of droplets on the targeted area via the method: the mass of each droplet is counted when it passes through this area. LWC, *u*, and *t* represent the liquid water content, the value of air speed in the far field and the tracking time respectively.

### 2.4. Film Model

The film model used in this code is proposed by Bai et al. [25], which can simulate the flow and heat transfer of the film above the ice surface. Some basic assumptions of this model are proposed as follows: (a) the film velocity normal to wall is zero; (b) gradients in wall-tangential direction are negligible compared to gradients in the wall-normal direction, which indicates that heat conduction and shear stress are dominant in the wall-normal direction. This assumption is applicable when the film thickness is very thin (far less than 1 mm). The dominant influence of this model in this direction is to transport the mass, momentum and energy. The film thickness, flow velocity and average enthalpy can be obtained by solving integral form of the transport equations from the surface of the airfoil to the top of the film in the normal direction.

The governing equations of the film motion are given:(6)$$\frac{\partial \rho \delta }{\partial t}+\nabla_{s}\cdot \left(\rho \delta U\right)=S_{\rho \delta ,imp}+S_{\rho \delta ,spl},$$
(7)$$\frac{\partial \rho \delta U}{\partial t}+\nabla_{s}\cdot \left(\rho \delta \mathrm{UU}\right)=-\delta \nabla_{s}\left(p_{imp}+p_{spl}\right)+S_{\rho \delta U,imp}+S_{\rho \delta U,spl},$$
(8)$$\frac{\partial \rho \delta h}{\partial t}+\nabla_{s}\cdot \left(\rho \delta Uh\right)=S_{\rho \delta h,imp}+S_{\rho \delta h,spl},$$
where δ and *h* represent the film thickness and enthalpy respectively. The subscript *s* in the $\nabla_{s}$ means that the discretization and equation solving is only performed on the extruded surface mesh. The right-hand-side terms of Equations (Equation 6)–(Equation 8) represent the source terms of mass, pressure, momentum and energy caused by droplets impingement and splashing, respectively. The detailed splashing model (terms with subscription spl) is proposed by Bai [25], which is already available in OpenFOAM. The mass source term $S_{\rho \delta ,imp}=\beta \cdot U_{\infty }\cdot LWC$ is the collected mass per unit time per unit area. The momentum source terms $p_{imp}$, $S_{\rho \delta U,imp}$ and the enthalpy source term $S_{\rho \delta h,imp}$ are associated with the mass source term $S_{\rho \delta ,imp}$. The corresponding velocity and temperature in estimating momentum and enthalpy is taken at impinging time. The momentum source is splitted into the tangential and wall-normal components, which are represented by $p_{imp}$ and $S_{\rho \delta U,imp}$ respectively.

The fundamental transport equations for the liquid film are solved on an extruded surface mesh. The extruded surface mesh is not necessarily connected to the airflow mesh in a geometric sense, as only boundary data exchange is needed between the two meshes. The mesh is discretized in both directions tangent to the surface, but is only one cell thick in the direction normal to the surface, as shown in Figure 3. This figure demonstrates the schematic diagram of data exchange between the film and air when solving Equations (Equation 6)–(Equation 8). In this figure, the thin solid lines represent the mesh around the airfoil surface, and the dotted line represents the single-layer virtual mesh for the film calculation.

Note that the water film is solved on a separate 2D mesh while the airflow is solved on a 3D mesh. A data exchange process is required to make the water film boundary conditions physically consistent. The top surface of the water film needs pressure, shear stress, heat flux information from the air field. Since the extrusion of mesh makes sure of face connectivity, thus a direct data mapping would suffice. The bottom surface of the water film uses a non-slip, fixed temperature boundary condition.

The gradients inside the film mesh are evaluated using least squares approach, similar to the one applied in flow simulation. The convection of velocity, turbulent kinetic energy and eddy frequency are discretized by upwind scheme to ensure boundedness.

### 2.5. Ice Accretion Model

#### 2.5.1. The Myers Model

The Myers model [13] is adopted to predict the icing rate on the airfoil surface. As is mentioned in Section 1, Myers model is more advantageous in glaze ice calculation since both the film flow on the airfoil/ice surface and the conduction between the ice and water layers are considered. At the initial stage, as the droplets impinge the airfoil surface, all of them freezes instantaneously into rime ice based on the assumption that water is not supposed to flow when the surface temperature is below the freezing point of water.

The heat flow balance equation is defined as
(9)$$Q_{k}+Q_{ai}+Q_{l}-Q_{ci}-Q_{d}-Q_{s}-Q_{cond}=0.$$
In Equation (Equation 9), $Q_{ci}$ represents the convective heat transfer between the ice layer and air, $Q_{s}$ represents the sublimation heat, $Q_{k}$ represents the kinetic energy of droplets, $Q_{l}$ represents the latent heat, $Q_{d}$ represents the sensible heat of droplets, $Q_{ai}$ represents the aerodynamic heat exerted on the ice surface by the air, and $Q_{cond}$ represents conductive heat inside the ice layer.

The ice growth rate equation is defined as
(10)$$\frac{\partial b}{\partial t}=\frac{{\dot{M}}_{collected}-{\dot{m}}_{s}}{\rho_{r}}.$$
In Equation (Equation 10), ${\dot{M}}_{collected}$ and ${\dot{m}}_{s}$ represent the collected mass per unit time per unit area of the incoming droplets and sublimation mass respectively.

In the second stage, water flow occurs when the latent heat is not enough to convert all the water into ice, which brings runback water, and the icing type changes from rime ice to glaze ice. The glaze ice conditions correspond to a regime between ice and water, and the temperature of both phases is supposed to be at the melting point. The reason for this two-stage process is that this model requires good heat transfer between the substrate and the incoming droplets. Under this prerequisite, the initial incoming water must immediately adopt the subfreezing substrate temperature and then freeze because it has a nucleation site. Only when there is a sufficiently insulating layer and enough energy has been introduced into the system can water appear. The formed layer may be relatively thin, but the freezing of even a small proportion of droplets may provide sufficient latent heat for the remainder to heat up and stay liquid as runback water. This is a consequence of an assumption that the droplets are supercooled and the temperature of the substrate remains fixed and subzero.

When the icing type turns to glaze ice, the heat flow balance equation is defined as
(11)$$Q_{k}+Q_{af}+Q_{l}+Q_{in}-Q_{cf}-Q_{d}-Q_{e}-Q_{cond}-Q_{out}=0.$$
In Equation (Equation 11), $Q_{af}$ and $Q_{cf}$ represent the aerodynamic heat exerted on the film surface by the air and the convective heat between the film surface and the air respectively. $Q_{e}$ represents the evaporative heat. The difference of $Q_{in}$ and $Q_{out}$ represents the energy that is introduced by the flux, the rest terms remain the same as the rime situation.

The ice growth rate equation is defined as follows:(12)$$\begin{array}{cc} \rho_{g}l_{f}\frac{\partial b}{\partial t} & =k_{i}\frac{T_{f}-T_{s}}{b}+k_{w}\frac{Q_{cf}+Q_{d}+Q_{e}-Q_{af}-Q_{k}}{k_{w}+\frac{Q_{cf}+Q_{d}+Q_{e}}{T_{f}-T_{a}}h}\\ \; & +Q_{out}-Q_{in} \end{array}$$
In Equation (Equation 12), $l_{f}$ represents the icing latent heat per unit mass, $k_{w}$ represents the thermal conductivity of water and $T_{a}$ represents the air temperature.

For a smooth transition from the rime to glaze models, the ice height, film thickness, and growth rate must be continuous, and thus $B_{g}$ and $t_{g}$ are introduced as the critical parameters to judge if rime ice changes to glaze ice. $B_{g}$ and $t_{g}$ are the thickness and the time at which glaze ice first appears respectively, which shows explicitly how they depends on the ambient conditions. $B_{g}$ and $t_{g}$ are defined as
(13)$$B_{g}=\frac{k_{i}\left(T_{f}-T_{s}\right)}{{\dot{M}}_{collected}l_{f}+Q_{a}f+Q_{k}-Q_{cf}-Q_{e}-Q_{d}},$$
(14)$$t_{g}=\frac{\rho_{r}B_{g}}{{\dot{M}}_{collected}}.$$
In Equation (Equation 13), $k_{i}$ represents the thermal conductivity of ice. $T_{f}$ and $T_{s}$ represent the melting temperature and wall temperature respectively.

As for the algorithm, the transition from rime ice accretion model to glaze ice accretion model is determined by the mass of the remaining water, which is obtained by the following equation:(15)$${\dot{M}}_{remain}={\dot{M}}_{collected}-{\dot{M}}_{rime}$$
In Equation (Equation 15), ${\dot{M}}_{remain}$ and ${\dot{M}}_{rime}$ represent the acculuated mass of the remaining water and rime ice per unit time, respectively. If the value of ${\dot{M}}_{remain}$ is greater than zero, the ice accretion model is switched from rime ice accretion model to glaze ice accretion model. $B_{g}$ is used to restrict the ice height for rime ice calculation, $t_{g}$ is used to determine when glaze ice is supposed to appear.

#### 2.5.2. Inner Loop of Ice Accretion

The ice accretion calculation is a multi-stage process with both outer loop (mentioned in Section 2) and inner loop calculation. The inner loop calculation is realized by splitting one outer step into several inner steps when it comes to calculating ice accretion and solving film equations, which is designed to predict the water flow more accurately since the glaze ice accretion is coupled with the film calculation. As shown in Figure 4, rime ice accretion is first solved in each inner step with the limitation of $B_{g}$. Glaze ice accretion occurs when there is accumulated water left. Furthermore, the film equations are solved after glaze ice accretion is calculated with the up-to-date source terms. The film thickness, film velocity and film temperature are obtained for the calculation of the next loop.

## 3. Validation on Airfoil Icing

Note that the solver is developed based on the unstructured finite volume framwork (OpenFOAM), which is naturally capable of simulating a 3D scenario. However, most experimental data available in the literature are airfoil icing measurement which are intrinsically quasi-2D or 2.5D, so the validation is limited to 2.5D in the current work. The purpose is to validate that the solver, developed on OpenFOAM, is capable of predicting accurate ice shapes.

### 3.1. Parameter Sensitivity Study for the Ice Shape

The mesh sensitivity study on the aerodynamic field is given in Section 2.1. In this section, the parameter sensitivity study on the ice shape is carried out using case 1 in Section 3.2. Figure 5 shows the predicted ice shapes using different numbers of inner loops and outer loops. The baseline configuration uses 6 outer loops (60 s/step) and 10 inner loops (6 s/step) for the total icing time of 360 s. The number of outer loops controls the frequency of aerodynamic and droplet trajectory evaluations. Increasing the number of outer loops to 9 (the 40 s/step) shows little effect on the ice shape while decreasing the number of outer loops to 4 (90 s/step) shows some difference. Increasing the number of inner loops reduces the time step size in evaluating the water film flow and ice accumulation. Doubling the number of inner loops shows a negligible effect on the final ice shape.

Figure 6 demonstrates the sensitivity of ice shape on the mesh resolution, where Figure 6a shows the effect of mesh coarsening. Baseline mesh is the one used in the later simulations. Specific zones are coarsened, by doubling the cell edge length, to show the mesh sensitivity. Coarsened mesh 1 is where the airfoil leading edge zone is coarsened. Coarsened mesh 2 is where coarsening is applied in the airfoil near-field based on coarsened mesh 1. Both show little effect on the ice shape, indicating that the mesh resolution near the airfoil is sufficient. Coarsened mesh 3 is where the total number of airfoil surface points is reduced by half with respect to the baseline mesh. In this case, only the ice height near the leading edge changes a little. Coarsened mesh 4 is where the airfoil near-field mesh is coarsened on top of 3.

Figure 6b shows the effect of mesh refinement. Specific zones are refined (reducing the cell edge length by half) to show the mesh sensitivity. Refined mesh 1 refers to the mesh with an additional Cartesian cut everywhere except for the leading edge refinement zone. Refined mesh 2 is where the leading edge zone is refined. Refined mesh 3 is the combination of 1 and 2. There is basically no influence on the final ice shape, which means the current airflow resolution, is sufficient for icing simulation. Note that earlier we have shown that the total number of airfoil surface points is dominating the final ice shape. Refinement mesh 4 is where the total number of airfoil surface points is doubled, which shows that the change on the ice shape is already small enough. The resulting ice shape suggests that the baseline mesh resolution has already reached mesh convergence.

The results in Figure 6 suggest that the current total number of points on the airfoil surface can not be further reduced; Mesh convergence has already achieved in the airfoil near field and the leading edge refinement zone. Given the observations above, the baseline configuration is selected for the validation cases in Section 3.2.

### 3.2. Analysis on the Results

The ability to accurately predict ice shape is validated on the NACA 0012 airfoil, with comparisons made to the ice shapes measured in experiments [26,27]. The geometry is a 2D NACA 0012 airfoil extruded in the spanwise direction. The visualization of the discretization on airfoil surface is shown in Figure 7. The sideview of this mesh is previously given in Figure 2. Note that the meshing tool snappyHexMesh only works for 3D mesh, the mesh on the leading edge is firstly refined in all directions to ensure the ability to capture ice growth accurately. Then an extrusion based on the side patch is required to generate a mesh that is homogeneous in the spanwise direction.

The size computational domain is 50c×50c×c, where *c* is the airfoil chord length. There are 50 layers in the spanwise direction. On the airfoil surface, the streamwise face length ranges from $2\times 10^{-3}c$ to $7.8\times 10^{-3}c$ depending on the location. The total cell size for the 2.5D cases is between 1.2 million and 1.5 million depending on different airfoils and ice shapes. The outer icing time step is set to 60 s for case 1 and 2, and 51 s for case 3. Within each icing time step, there are 10 inner steps to march the water film and accumulate ice. The final ice shape does not show significant sensitivity to the time steps size.

The results are then calculated using both the Eulerian and the Lagrangian methods. Different types of ice shapes are calculated including rime ice and glaze ice with small-size droplets, and SLD ice accretion.

The icing simulation conditions for the small-size droplets and SLDs are shown in Table 2. Note that for cases 1 and 2, the droplet size is small, thus the Eulerian method is used. For case 3, the Lagrangian method is use to track the SLDs.

The collection efficiency for case 1 is shown in Figure 8. In the Eulerian computation, the collection efficiency on the airfoil surface is distributed mainly around the leading edge as expected. More importantly, the distribution shows a 2D distribution as the collection efficiency distribution is homogeneous in the spanwise direction. It is expected since the Eulerian method treats the droplets as a continuous phase.

The validation results of ice shape for cases 1 and 2 are shown in Figure 9 and Figure 10. The results calculated with the Eulerian method show the homogeneity of ice shape along the spanwise direction, which is a nature consequence of the previously mentioned distribution of collection efficiency. The 2D ice shape fits well with the experimental results, which proves that the tracking and collecting method, the film model and the ice accretion model are feasible in predicting ice shapes under small-size droplet conditions.

For case 1, the upper and lower limits where the droplets hit agrees with the experiment result well. However, for case 2, the upper limit is closer to the leading edge compared to the experiment. It is possible that the error is originated from the airflow computation considering the ice shape causes flow separation on the upper surface. The steady-state solution from the RANS equation is known to overpredict separation bubble size. For that reason, given the high efficiency of RANS, it is acceptable for engineering purposes.

Different from cases 1 and 2, the droplet size in case 3 belongs to the SLD category. Since the large droplets (SLDs) are less uniform and are more likely to splash, the resulting distribution of collection efficiency should be less uniform than small droplets. Thus the Lagrangian method may be more appropriate than the Eulerian method. In our numerical simulations of case 3 using the Lagrangian method, the injection positions of the large droplets are randomly determined at the inflow plane. The distribution of collection efficiency on the airfoil surface is shown in Figure 11. The discontinuity of the distribution demonstrates the advantages of the Lagrangian method for tracking SLDs.

The global view of ice shape and comparison with experiment measurement of case 3 are shown in Figure 12. The result calculated with the Lagrangian method demonstrates some ice roughness, which is caused by the nonuniform distribution of the collection efficiency shown in Figure 11. This nonuniform distribution is owing to the distinct feature of the Lagrangian method in presenting the sparsity and randomness of the large droplets. This may serve as a tool to help further get down to the real ice roughness, which will not be discussed in the current scope.

The spanwise-averaged 2D ice shape (Figure 12b) implies that the collection efficiency is slightly larger than the experimental one. Since the main feature of the ice shape is captured, it is still reasonable considering the difficulty in numerical prediction SLD icing and the complex physics associated with SLDs. It proves that the icing simulation method is also feasible in predicting ice shapes under SLD conditions with the Lagrangian method, but the accuracy remains to be improved.

In summary, the icing simulation proves to be feasible under small-size droplet condition with the Eulerian method and SLD condition with the Lagrangian method. Characteristics of these two numerical methods of tracking and collecting droplets are demonstrated respectively. For the Eulerian method, the collection efficiency is calculated as a field, which is continuous in time and space, and this brings a clean ice shape as a result. This can be quite useful and time-saving when the body shape is complex. For the Lagrangian method, it is consistent that the droplet injection takes the randomness of droplets’ initial positions into account and the ice shape is more close to the experimental one with ice roughness generated, but this is time-consuming when the body shape is complex.

## 4. M6 Wing Icing Simulation

Although there is a lack of trustworthy experimental data on realistic 3D geometries, the full 3D capability of the icing simulation code can still be demonstrated. The ONERA-M6 wing is studied in icing conditions. This wing is selected because it is a well-known geometry with several available experimental and numerical results for the airflow solution. It is often used for validation of numerical results in the transonic range, at high Mach number and high Reynolds number. In the context of ice accretion simulation, the Mach number and the Reynolds number have therefore been reduced.

The calculation conditions are shown in Table 3. The computational domain is about 40c×40c×25c where *c* is the chord length at wing root. Extra refinement of the volume mesh within the distance of *c* to the wing surface is applied. For this case, the number of cells is 1,630,000. The air flow field and ice shape does not show significant sensitivity to mesh resolution at this level. The mesh on the leading edge are refined to capture the 3D ice shape, as shown in Figure 13.

The predicted ice accretion on the M6 wing is shown in Figure 14. Figure 14b shows a zoomed-in view on the ice shape. The global view shows that most of the water droplets hit the region near the leading edge. From the zoomed-in view, it is obvious that ice shape vary significantly in the spanwise direction. It is mainly due to the swept effect of the wing, which causes water film to flow toward the wing tip and cause the wavy ice shape along the spanwise direction. Qualitatively, the wavy ice shape along the spanwise direction is consistent with what is observed in the experiment and flight test. Further work in the future is needed to evaluate how accurate the current approach can give in a real-world 3D scenario. This roughness is caused by the 3D effects, which is more prominent than that of a 2.5D case. Furthermore, slices are extracted from different positions along the spanwise direction to analyze the tendency of the change of ice shapes. It can be deduced from Figure 15 that as the spanwise position of the slice goes from the wing root to the wing tip, the ice shape tends to show more features of ice angles, which means that glaze ice first appears at the tip. Furthermore, the formation of ice angles is related to some factors including the figure and the size of the airfoil, which may be discussed further in future work.

This case just proves the 3D icing simulation ability. Further complex 3D research can be performed via this code.

## 5. Comments on the Solver Efficiency

In the above, the viability of developing a 3D icing solver based on the open-source platform OpenFOAM is demonstrated. The bottleneck of solver robustness, which is the mesh quality for 3D icing simulation, is circumvented by using the mixed Cartesian/body-fitted meshing tool (snappyHexMesh) which can generate reasonable mesh on complex icing shapes and still preserve the ice shape. Another important aspect, of a numerical solver, is speed. Here the solver speed at different simulation scenarios are presented.

The meshing parameters are carefully adjusted to find a balance between efficiency and accuracy. Although we did not show a pure 2D case in the above sections (just 2.5D with one spanwise layer), it can serve as a good reference for speed estimation. Generally for 2D simulations, the Eulerian method can return a prediction of 360 s of icing time (seven outer icing time steps) around 500 s wall clock on eight Xeon cores. The bottleneck is at the meshing step using the snappyHexMesh. It is because the snappyHexMesh can only generate a 3D mesh. An extrusion process is still required to generate a mesh that has only one layer in the spanwise direction. The Lagrange method may take up to 900 s for the same case. The additional time is spent on particle tracking where load balancing between cores becomes an issue. However, this is not the focus of the current work.

For the 2.5D simulations shown in Section 3, the time spent on the meshing process is almost the same, just with more extrusion layers along the spanwise direction. More computational resources are spent on solving PDEs and particle tracking ODES. The wall clock time is about 20 min and 45 min for the Eulerian and the Lagrangian simulations in Section 3 on eight Xeon processors, respectively.

For the M6 wing simulation, the issue of uneven load balancing for the particle tracking is more severe, and the generation of a high-quality true-3D mesh certainly requires more time. It takes about 4–18 h depending on the Eulerian/Lagrangian method selected.

## 6. Conclusions

This paper introduces a 3D ice simulation code that is developed and implemented in OpenFOAM. Some utilities and solvers already present in OpenFOAM are used and combined to realize the icing simulation. Validations in 2.5D are performed to prove the feasibility and accuracy of the icing simulation ability. It is worth noticing that both the Eulerian method and the Lagrangian method are applicable in the code. For small-sized droplets, the Eulerian method is used. A volumetric fraction is solved with the airflow to replace the droplet tracking process, which is advantageous in solving small-size droplet cases with complex 3D geometries. For SLDs, Lagrangian method is used. Droplets are tracked at individual parcels. The physical characteristics of SLDs can be clearly described, such as splashing or bouncing with the Lagrangian method, which is not investigated much further in this paper. The ice roughness is apparently calculated as a result, which also remains to be further investigated. Finally, a test on the 3D M6 wing is performed to demonstrate the full 3D calculation ability.

Much work still needs to be done to develop and improve the code. Besides the code itself, it is worthy of attention that droplet size distribution affects the ice shape greatly, especially the SLD conditions that cover a wide range of the droplet sizes. In this paper, methods of tracking and collecting droplets are separated and the median volume diameter is used as the fixed droplet size instead of the sizes with a distribution. The next step is to make a combination of the Eulerian and the Lagrangian methods to treat the droplets with the given distributed sizes to realize the ability to simulate the ice accretion in the real SLD environments.

## Figures and Tables

**Figure 1 entropy-24-01365-f001:**
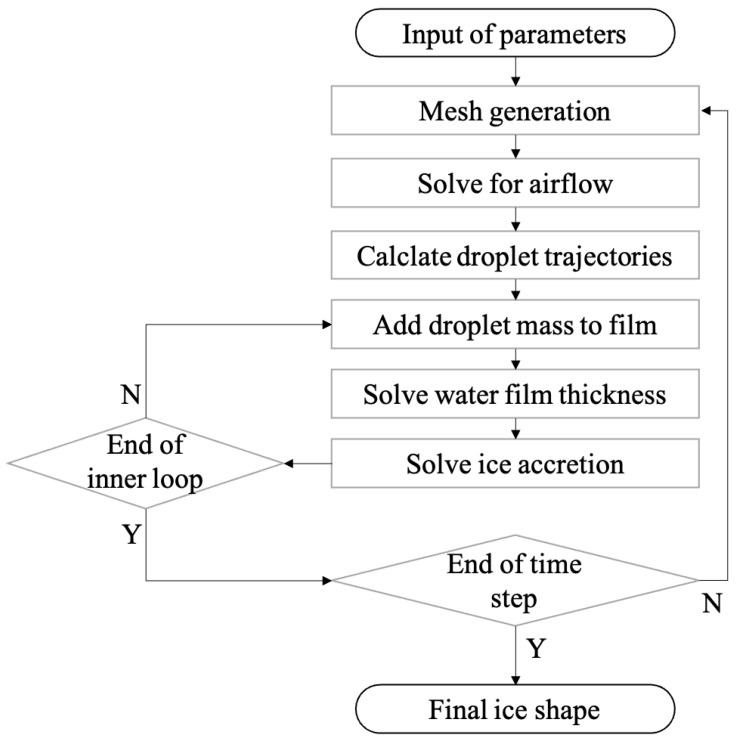
Diagram of the present icing simulation process.

**Figure 2 entropy-24-01365-f002:**
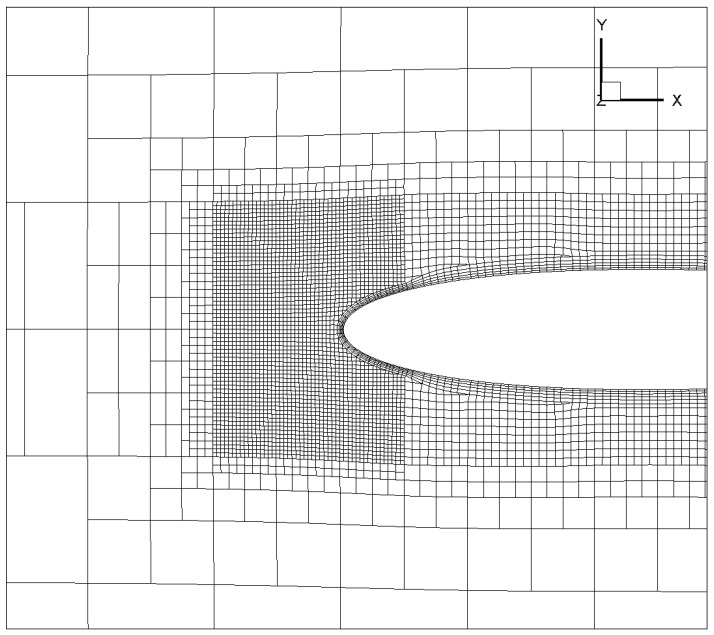
Spanwise view of the mixed Cartesian/body fitted mesh generated for the present airflow simulation.

**Figure 3 entropy-24-01365-f003:**
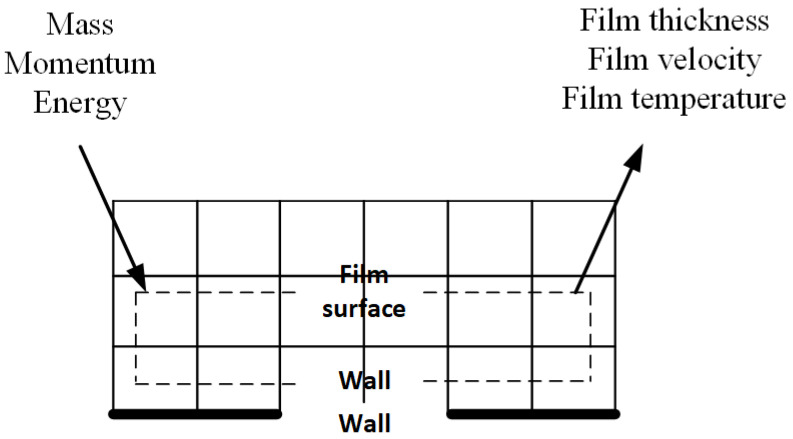
Diagram of data exchange between airflow and film on the virtual extruded mesh.

**Figure 4 entropy-24-01365-f004:**
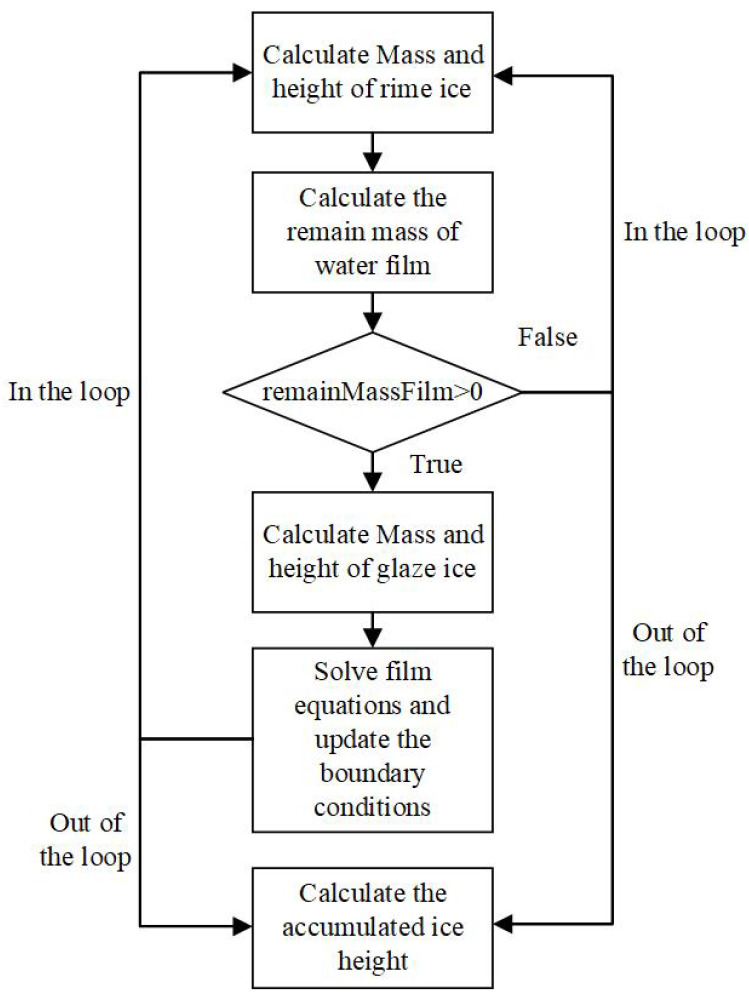
Diagram of inner-loop ice accretion method.

**Figure 5 entropy-24-01365-f005:**
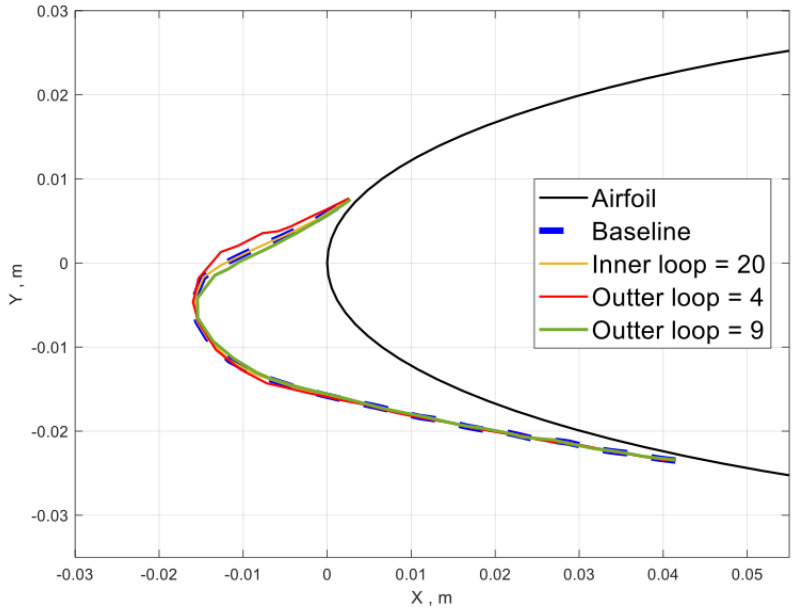
Ice shape sensitivity study on number of solver steps.

**Figure 6 entropy-24-01365-f006:**
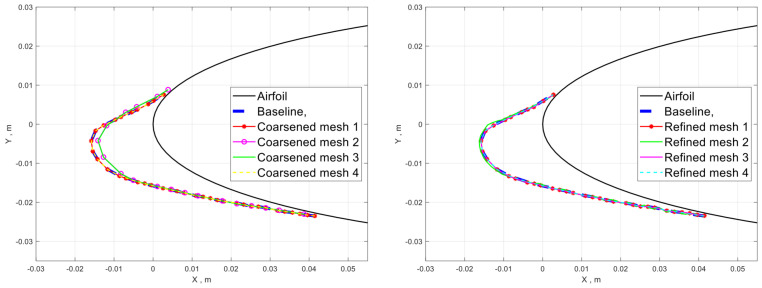
Ice shape sensitivity study on the meshing parameters.

**Figure 7 entropy-24-01365-f007:**
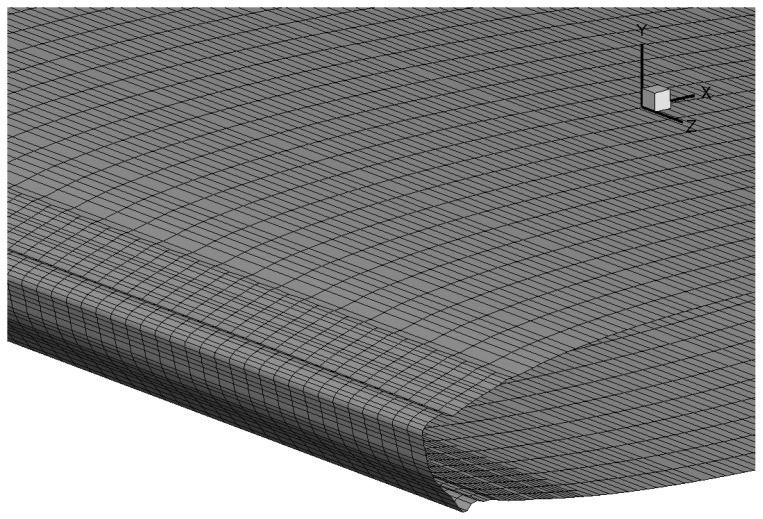
Mesh on the airfoil surface for 2.5D icing simulation on NACA 0012.

**Figure 8 entropy-24-01365-f008:**
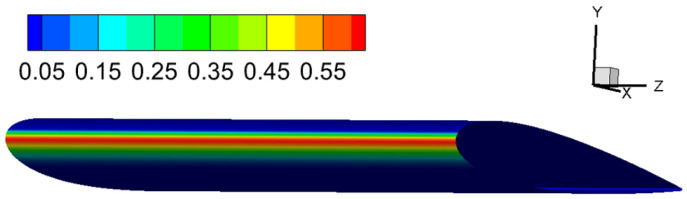
Case 1: distribution collection efficiency predicted by the Eulerian method.

**Figure 9 entropy-24-01365-f009:**
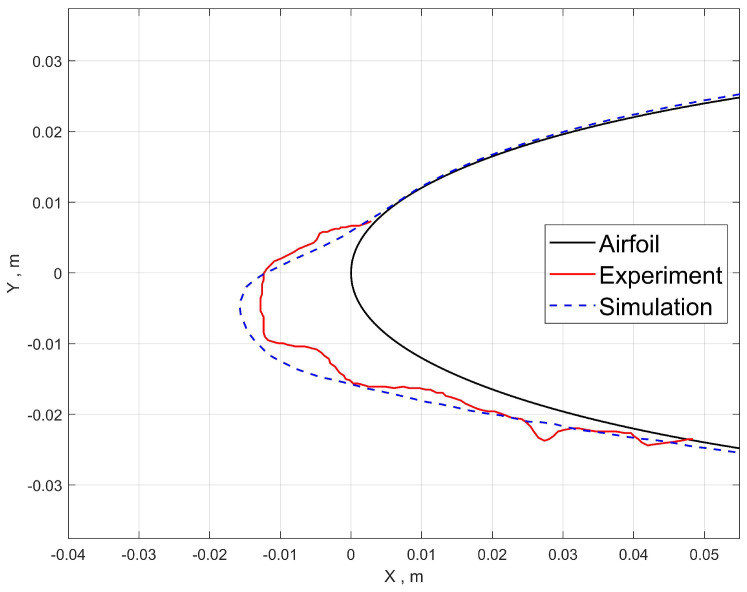
Case 1: predicted ice shape compared to experiment.

**Figure 10 entropy-24-01365-f010:**
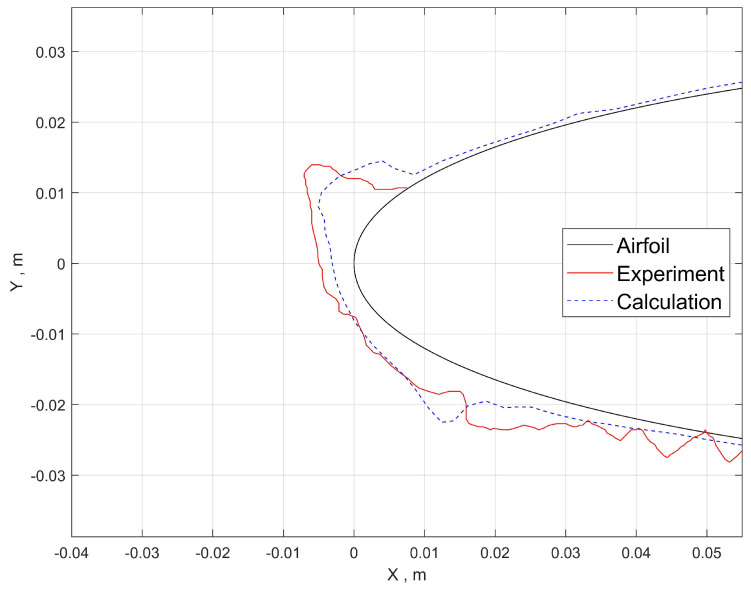
Case 2: predicted ice shape compared to experiment.

**Figure 11 entropy-24-01365-f011:**
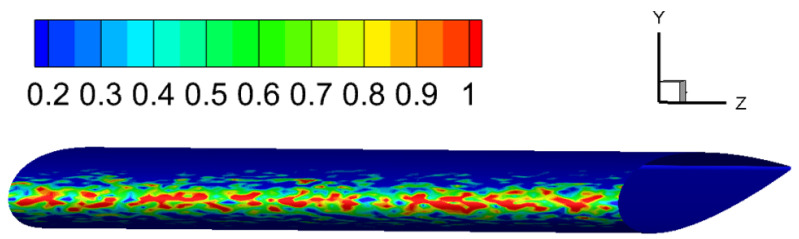
Case 3: distribution collection efficiency predicted by the Lagrange method.

**Figure 12 entropy-24-01365-f012:**
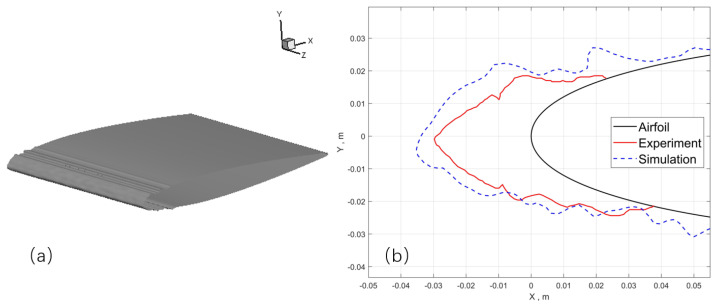
Case 3: (**a**) global view of the predicted airfoil with ice; (**b**) Spanwise-averaged ice shape compared to experiment.

**Figure 13 entropy-24-01365-f013:**
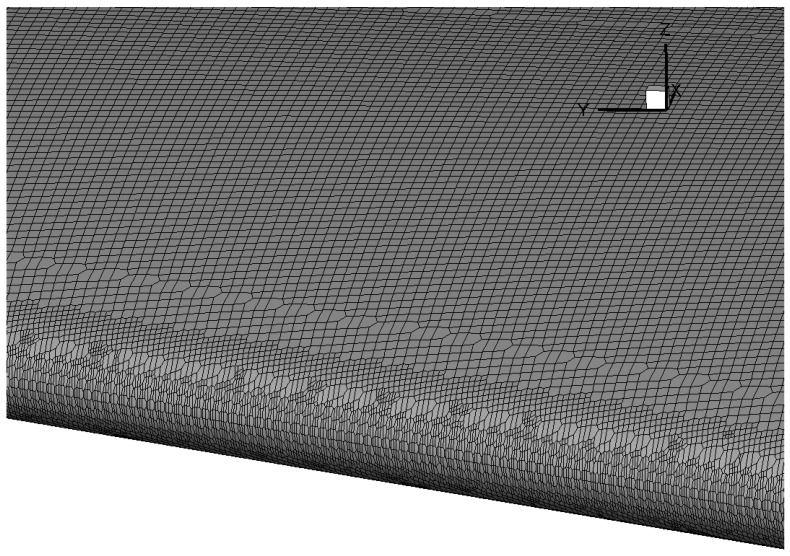
The surface mesh near the leading edge of the M6 wing.

**Figure 14 entropy-24-01365-f014:**
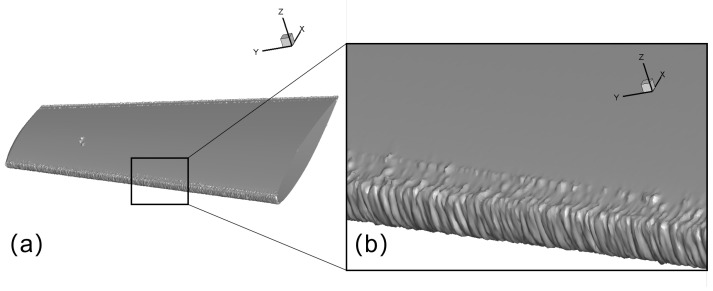
Predicted ice accretion on M6 wing (**a**) global view, (**b**) zoomed-in view.

**Figure 15 entropy-24-01365-f015:**
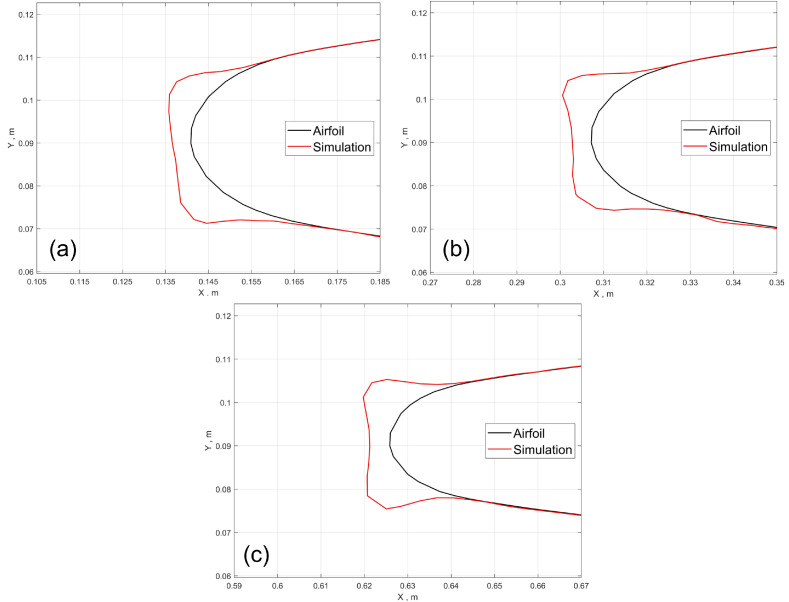
Predicted ice shapes for the M6 case at different spanwise locations from wing root to wing tip: (**a**) 20%, (**b**) 44%, (**c**) 90%.

**Table 1 entropy-24-01365-t001:** The tested parameters and their ranges in the mesh sensitivity study.

Parameter Name	Lower Bound	Upper Bound	Resulting $C_{l}$ Error
Cartesian refinement level	3	7	0.2–3.6%
Size of refinement zone	1.5c×0.5c	12c×6c	1.9–2.1%
Initial mesh resolution	45×25	350×350	0.9–3.7%
No. wall layers	7	10	0.5–1.9%
Farfield distance	10c	50c	1.1–1.7%

**Table 2 entropy-24-01365-t002:** Calculation conditions for the validation cases of NACA 0012.

	t (s)	T (°C)	V (m/s)	AOA (°)	LWC (g/m${}^{3}$)	D (μm)
Case 1	360	−28.3	67.05	4	1	20
Case 2	360	−4.4	67.05	4	1	20
Case 3	714	−19.2	77	0	0.65	70

**Table 3 entropy-24-01365-t003:** Conditions for M6 wing.

	t (s)	T (°C)	Ma	V (m/s)	AOA (°)	LWC (g/m${}^{3}$)	D (μm)	Re (${}^{-1}$)
Case 4	420	−7	0.367	120	0	1	20	$8.3\times 10^{6}$

## Data Availability

The data presented in this study are available on request from the corresponding author.

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
