# Peer review of "Development of a 3D Eulerian/Lagrangian Aircraft Icing Simulation Solver Based on OpenFOAM"

_entropy, 2022, doi:10.3390/e24101365_

Round 1
Reviewer 1 Report
The article titled "Development of an OpenFOAM-based 3D Eulerian/Lagrangian aircraft icing simulation solver", written by Han Han, Zifei Yin, Yijun Ning and Hong Liu, is about a simulation method for the formation of ice on aircraft This important issue for flight safety has been addressed by the CFD community for several decades with a certain degree of maturity, evolving at the same pace as both hardware and numerical methods.
The article presents some minor novelties such as the combination of the Eulerian and Lagrangian formulation to treat the entire set of drop sizes, those smaller with the first and the largest with the second. It also adds a 2D surface representation of the water film that forms once the ice dries, giving the phenomenon more physical realism. For the formation of the frost layer and glazed ice, the authors have chosen the Myers model, a well-known improvement on the Messinger model. For air, the authors state to use a compressible semi-implicit method for pressure-bound equations with RANS turbulence model.
Although the article is interesting, quite well organized and written, there are some details that deserve to be modified, corrected or added before it is ready for publication, always thinking of improving its quality both on the side of the reader and the writer.
In general, the main criticisms are the lacking of a strong demonstration that 3D results are really good. Even though the lack of enough experimental or virtual results for comparisons there is some basic proofs that deserve to be done in order to be sure about the accuracy of the results, among them a mesh convergence analysis. Also the inclusion of Cd and Cl with and without ice layer is very important for comparison against another works.
Another thing that is important to fix is the incomplete definition of most of the models used.
Also it is important to include the computational cost involved.
These things should be included in the revised version and the conclusions reviewed with this extra information.
In the following I cite most of them, probably there are another, in particular in the grammatical sense.
-
In Table I, I wonder if the film model is missing in the diagram. Is that right ? If yes please include it.
-
Line 2 and 6
-
Cartesian instead of Cartisian,
-
,Two with lowercase and
-
tracking methods are adopted instead of is adopted.
-
Before Equation (1) non-dimensional or dimensionless instead of non-demensional
-
In equations (1) and (2),
-
If LWC is the liquid water content and rho_d is the droplet density (water) I don't understand how alpha is dimensionless ? I think that the confusion is about that LWC is not the water content in the sense of mass or volume else in the sense of density. So, you should fix this putting density instead of water content.
-
On the other hand if this section deals with the Eulerian part of the droplets dynamics, you should mention that this applies only for small droplets, and also that LWC should be only the density of small water droplets (d < 50 microns). Is it right ? If yes, please fix the definition of LWC saying that is the density of the smaller portion of water droplets.
-
Moreover, assuming alpha and Ud dimensionless in Eq(1), so, t and nabla should also be dimensionless !!!. Please clarify this fact !
-
On the other hand I consider not much clear the mixing between dimensionless variables with another variable with dimensions. The role of Froude and Reynolds number is currently well understood and it is not necessary to put explicitly in Eq (2). So I recommend to present the equations in variables with dimensions to avoid errors.
-
-
In Eq (2) the non-conservative form for the convection term implies that Ud is divergence free. Using OpenFOAM it is more straightforward to use the conservative form including U square inside the divergence operator and avoiding the above assumption, i.e. Ud as a solenoidal field.
-
Also in Eq(2) the bold "g" should be presented.
-
Thinking about Eqs (1) & (2) representing the small droplets modeled in Eulerian form, in conjunction with the airflow representing an Euler-Euler (two fluid model) where you can add more physics to the interaction between both phases. For example the turbulent dispersion at least in one way, using the turbulent kinetic energy and the eddy viscosity of turbulence modeling to produce some mixing in the small droplets domain like a diffusion term. So, not for the present model but for the future you can think about adding some diffusion term coming from the airflow turbulent fields.
-
Equation 3: beta is dimensionless ? Do you use Un as dimensionless also ? Clarify
-
Line 156, What kind of interpolation do you use ?. Please clarify in the text some brief details about this interpolation method. Have you used barycentric or area coordinates.
-
In the film model section, when you say “ This assumption is applicable when the film thickness is very thin (far less than 1mm). “ I wonder if this assumption does not depend on the relative thickness respect to the cell normal size. Please if this comment is correct modify the text accordingly
-
After Eqs (6) to (8), I recommend to include the expressions of the source terms as you have implemented in order a user may reproduce your results.
-
In equations (6),(7) and (8) the \nabla_s (gradient operator) is not defined. Neither the subindex of the two last momentum sources. I wonder if the subindex are typed in a right way or if there is some typo error !
-
Line 176 to 178, It is not clear neither the interpolation procedure between both meshes to share information needed by each flow domain nor the boundary conditions applied to the film model equations ? I understand that this film model works like a shallow water model in the sense that the variables are averaged in the film height neglecting its thickness and the solver gives the variation in the tangential manifold. But this domain, the manifold, need boundary conditions for solving Eqs (6),(7) and (8), not enough clearly defined. Please include the boundary conditions for the three variables, delta, U and h on the film model. Also it should be more clear if you add a figure showing the airfoil and the film manifold and where the boundary conditions are specified and the interpolation between both meshes is carried out.
-
At the beginning of Myers model section, I consider that the shear stress is another important fact to determine if the rime ice stick on the airfoil surface or if it is drift by the airflow. Is that right ? If yes, how do you justify to neglect this effect ?
-
In Eq (10), Please clarify the units involved in this expression. For me it is not clear because b has length units, t is time, M is mass and rho is density.
-
Line 201 and the following, Why do you change B from lowercase to uppercase ? For not confusing the reader I prefer that you keep the lowercase for the ice layer height. I suppose that this change respond to the original paper of Myers, but here it is not necessary to do this change.
-
In the validation section, In this publication there is several results for comparison, see Wright, W. B., and Rutkowski, A., “Validation Results for LEWICE 2.0” [CD-ROM], NASA CR 1999208690, Jan. 1999.
-
Line 282-283, I think you need to refine the mesh, particularly in the icing areas to show that your results have converged on the mesh, as a starting point to be able to say that the results are acceptable, especially when you don't count with benchmark results.
-
Line 296-297, I recommend to see the article: "Hann, Richard; Hearst, R. Jason; Sætran, Lars Roar; Bracchi, Tania (2020). Experimental and Numerical Icing Penalties of an S826 Airfoil at Low Reynolds Numbers. Aerospace, 7(4), 46–. doi:10.3390/aerospace7040046 " for a challenge validation
Reviewer 2 Report
The present paper provides an implementation of aircraft icing modeling techniques within the OpenFOAM open-source CFD framework. The content of the paper is within the scope of the Entropy Special Isssue Applications of CFD in Heat and Fluid Flow Processes and thus it can be considered for publication.
Prior to acceptance, the following minor issues should be addressed by the authors (please check also ALL the comments within the annotated pdf document):
1 1) typos and text corrections/inconsistencies: please double-check the text for small errors and inconsistencies (see also the comments/deletions within the annotated pdf document);
2 2) details on the numerical setup for the validation and 3D-test cases (see also comments within the annotated pdf document): the authors must include full details on the computational setup for each case, including domain geometry/dimensions, boundary and initial conditions, solution algorithm (steady vs unsteady), discretization schemes, mesh characteristics and mesh independency tests (if any).

Reviewer 3 Report
General comment
The manuscript “Research on wind load characteristics on the surface of a towering precast television tower without building envelope” has been reviewed in detail. The study deals with a very interesting and peculiar topic previously investigated by other scientists. The manuscript is well-organized and well-written, the text is fluent and comprehensive both for expert and no-expert people. In general, the reviewer believes that the present manuscript might be published after addressing some comments and suggestions provided below.
Abstract
At line 6: “… , two droplets …”
1. Introduction
In general, could the authors explain in this section, or maybe in another one, why the Lagrangian approach would not be applicable/feasible for the present case study? The reviewer agrees with the authors on the very expensive computational nature of this numerical approach, but here the case study is not that big.
At line 3: “… phenomenon [1]”. This mistake “text[xx]” has been found very often throughout the whole manuscript.
At line 29: “wind-tunnel tests”, please consistent throughout the whole manuscript.
At lines 32-34: “… (such as …), because …”.
At line 39: mesh or grid, but not both, please choose one and stick with that throughout the whole manuscript.
At lines 64-65: could the authors elaborate a bit more about the goal and novelty of the present study? Moreover, could they clarify the statement “advantageous in mesh generation”? Do they refer to the “SnappyHexMesh tool” or anther tool developed by them? Finally, SnappyHexMesh can be faster than other meshing software and for that convenient, but it could also be way less accurate, as well-known.
At lines 66-69: “Section X” and not “Section. X”. The same mistake has been identified also with “Figure …”. Please apply this correction whenever necessary throughout the whole manuscript.
2. Numerical method of icing simulation
It would be good to make text and figure consistent by using the same keywords used in the figure.
2.1. Mesh generation
Could the authors provide some specifications about the grid size (min, max size close and far from the airfoil) as well as another view of the grid as zoom out?
Could the authors explain if a grid-sensitivity analysis has been carried out? And if not, why?
At line 86: “a generally reduction of …”, unless you mention what “much reduction” means quantitatively.
Figure 2: is this a figure from the present case study? Could you please make the caption of the figure a little bit more exhaustive?
At line 104: please provide first the full-extensive name, then the acronym “SST k-ω model”. On this subject, could the authors carefully explain why the present turbulence model has been selected and if any other turbulence model (available in OpenFOAM) has been considered at least for a sensitivity investigation?
At lines 104-105: could the authors provide a range of the y+ parameter for such kind of simulations, please?
Could the authors mention the residual threshold to assume the simulations converged?
2.3.1. Eulerian method
The sentence is a bit vague and not necessarily true “Thus the Eulerian method is more suitable in calculating the collection of small size droplets”. What do authors mean with “small size”?
3. Validation
At lines 219-220: the sentence “2.5D …[24,25]” is unclear. What did you exactly validate and how?
At line 221: the statement “when snappyHexMesh works” is unclear.
At line 221: the statement “The effects of the 2.5D grids … ” on what? This is also unclear.
In general, as commented above, more detailed views about the computational grid must be shown either here or in previous section.
At line 224: what do authors mean with “2D conditions”? This is unclear and too vague, please rephrase here.
At lines 228-231: the sentence “For Eulerian … in Figure 6” is unclear and also redundant, it should be rephrased.
Figure 6: could the authors optimize the space in this figure by re-distributing the position of the colorbar (legend) and coordinate system?
Table 1: in the caption please try to be more detailed by specifying the calculation conditions.
Figures 7-8: these two figures could be re-grouped in Fig. 7 only;
Figure 9: same comment of Figure 6.
At line 249: could the authors explain what they mean with “but still the result is acceptable”? What is the threshold of acceptance?
Figure 10: could the authors place these two plots one close to each other (right-left)?
4. M6 wing icing simulation
At line 262-263: the statement “the demonstrate the full 3D capability of the icing simulation code can still be demonstrated” is redundant and unclear.
At line 268: could you please explicitly indicate the Mach and Reynolds numbers of this study, please?
Figure 11: as stressed also above, the caption must be more descriptive so that the figure can standalone.
Figure 12: could the authors change the style to show the accretion? This is not very comprehensive in the present form. It would be good to use a colorbar also here. And as commented in Figure 10, could you place these two plots one close to each other (right, left)?
Figure 13:
1) it is not clear the correlation between this figure and Figure 12. Could you please report the axes also on Figure 12?
2) as in Figure 10, could re-arrange these to fit into one line please?
Round 2
Reviewer 1 Report
Even though I recognize the effort done by the authors to improve the article, at this point it is still unacceptable to say that the convergence of the mesh is obvious. It's not a good answer so you can convince me of it. If it is unaffordable for the authors to do further refinements, I would probably be convinced to show the convergence of Cd, Cl and the shape of the ice accretion by thickening the mesh, using for example a mesh with sqrt(2) times the current size and then another with mesh size doubled to give me tell how convergence is. achieved and why this current mesh is good enough.
Author Response
The authors thank the reviewer for his/her honesty and responsible attitude to our manuscript. Indeed, this is what makes the manuscript better.
We dug back to a development log on snappyHexMesh parameters in March 2020 done by the first author in his first year of graduate study. Now we added that part of the sensitivity study to section 2.1.
We also added a sensitivity study on the ice shape to section 3.1.
All the changes are marked in red. Again, thank you for being patient and responsible as a reviewer.
Reviewer 3 Report
The authors addressed all the comments provided by reviewer and the manuscript is now ready for publication.
Author Response
Thanks very much for the reviewer's help in improving the manuscript.
Round 3
Reviewer 1 Report
I accept the minor changes made in the manuscript without respecting my request to include at least the convergence curves of the CL, assuming that the focus of the paper is on the development of the ice sheet. I suggest the authors and the editor of the journal to be more respectful of the reviewer's time and work in the future and take time (not hours) to respond accordingly. It seems to be a quick race to publish, when in science knowledge must be strongly supported.
Author Response
Thank you for your approval and we appreciate the time and effort you have spent improving our manuscript.